# Statins and renal disease progression, ophthalmic manifestations, and neurological manifestations in veterans with diabetes: A retrospective cohort study

Ishak A. Mansi[1,2,3]*, Matheu Chansard[4], Ildiko Lingvay[3,5], Song Zhang[5], Ethan A. Halm[6], Carlos A. Alvarez[5,7]

1 Department of Education, Orlando VA Healthcare System, Orlando, Florida, United States of America, 2 Department of Internal Medicine, University of Central Florida, Orlando, Florida, United States of America, 3 Department of Internal Medicine, University of Texas Southwestern Medical Center, Dallas, Texas, United States of America, 4 Department of Anesthesiology and Pain Management, University of Texas Southwestern Medical Center, Dallas, Texas, United States of America, 5 Department of Population and Data Sciences, University of Texas Southwestern Medical Center, Dallas, Texas, United States of America, 6 Department of medicine, Robert Wood Johnson Medical School, New Brunswick, NJ, United States of America, 7 Department of Pharmacy Practice, Texas Tech University Health Sciences Center, Dallas, Texas, United States of America

* ishak.Mansi@va.gov

## Abstract

### Background

Statins increase insulin resistance, which may increase risk of diabetic microvascular complications. Little is known about the impact of statins on renal, ophthalmologic, and neurologic complications of diabetes in practice. The objective of this study was to examine the association of statins with renal disease progression, ophthalmic manifestations, and neurological manifestations in diabetes.

### Methods

This is a retrospective cohort study, new-user active comparator design, that included a national Veterans Health Administration (VA) patients with diabetes from 2003 to 2015. Patients were age 30 years or older and were regular users of the VA with data encompassing clinical encounters, demographics, vital signs, laboratory tests, and medications. Patients were divided into statin users or nonusers (active comparators). Statin users initiated statins and nonusers initiated H2-blockers or proton pump-inhibitors (H2-PPI) as an active comparator. Study outcomes were: 1) Composite renal disease progression outcome; 2) Incident diabetes with ophthalmic manifestations; and 3) Incident diabetes with neurological manifestations.

### Results

Out of 705,774 eligible patients, we propensity score matched 81,146 pairs of statin users and active comparators. Over a mean (standard deviation) of follow up duration of 4.8 (3)

**Data Availability Statement:** Data cannot be shared publicly because of VA confidentiality and patients protections rules. Data are available from

the VINCI Institutional Data Access for researchers who meet the criteria for access to confidential data (www.VINCI.va.gov).

**Funding:** The author(s) received no specific funding for this work.

**Competing interests:** NO authors have competing interests.

years, renal disease progression occurred in 9.5% of statin users vs 8.3% of nonusers (odds ratio [OR]: 1.16; 95% confidence interval [95%CI]: 1.12–1.20), incident ophthalmic manifestations in 2.7% of statin users vs 2.0% of nonusers (OR: 1.35, 95%CI:1.27–1.44), and incident neurological manifestations in 6.7% of statin users vs 5.7% of nonusers (OR: 1.19, 95%CI:1.15–1.25). Secondary, sensitivity, and post-hoc analyses were consistent and demonstrated highest risks among the healthier subgroup and those with intensive lowering of LDL-cholesterol.

## Conclusions

Statin use in patients with diabetes was associated with modestly higher risk of renal disease progression, incident ophthalmic, and neurological manifestations. More research is needed to assess the overall harm/benefit balance for statins in the lower risk populations with diabetes and those who receive intensive statin therapy.

## Introduction

Type 2 diabetes mellitus has been considered a "cardiovascular risk equivalent" [1], resulting in a universal recommendation of statins for all patients with diabetes aged 40 to 75 with LDL-cholesterol 70 to 189 mg/dL for primary prevention of cardiovascular diseases (CVD) [2]. Despite their cardiovascular benefits, statins have also been shown to increase insulin resistance [3–8], which is thought to be a main driver of the pathogenesis of diabetic microvascular complications [9–13].

There is a paucity of data on the effects of statins on diabetic microvascular complications. The landmark cardiovascular randomized controlled trials (RCTs) that support the current guidelines for statin use for primary prevention did not, *a-priori*, evaluate their potential impact on diabetic microvascular complications. A handful of observational studies reported an increased risk of diabetes microvascular complications associated with statin use [14–17], of which two studies were significantly larger (60,455 patients followed for a mean of 4 years and 25,970 patients followed for 6.4 years) [14, 15] than RCTs which established statins' safety. However, other observational studies found no association between statin use and increased risk of diabetic neuropathy and/or diabetic retinopathy [18–20] and two small trials (less than 50 participants each) associated statin use with improvement in diabetic retinopathy [21, 22].

Reasons for these conflicting results include inadequate adjustment for baseline confounders and the short duration of follow up. On one hand, statin use may be falsely associated with better outcomes because of healthy-user bias, and being a surrogate for higher quality of care, or better access to healthcare [23, 24]. Alternatively, statin use may be falsely associated with worse outcomes because of more exposure to healthcare resulting in ascertainment bias or confounding by indication [24].

The objective of this study was to address these methodological concerns by employing a new user design with active comparators to examine the association of statin therapy with incidence of renal disease progression, and diabetes with ophthalmic and neurological manifestations in a large national cohort of patients with diabetes in the Veterans Affairs (VA) health system who had significant longitudinal follow-up and who have detailed data on healthcare utilization, medical encounters, medication history, vital signs, and laboratory investigations to minimize confounding.

## Methods

### Study design

This is a retrospective cohort study using the national VA Corporate Data Warehouse (CDW), which encompasses inpatient and outpatient diagnosis/procedure codes, pharmacy, vital signs, and laboratory data. CDW catalogues its data according to published protocols (S1 File) [25]. This study cohort was assembled from a national VA cohort with diabetes identified using a validated algorithm [26] that has been described previously [27]. Briefly, we assembled a cohort of statin users and nonusers (overall cohort), aged 30 years or older, and who are regular VA users. We defined regular VA users as having all of the followings during each of the baseline and the follow-up periods: 1) at least one VA health care encounter; 2) blood pressure and weight measurements; 3) pharmacy records of medications; and 4) laboratory data that included blood/serum glucose, creatinine, and LDL-cholesterol. Available data for included patients encompassed all encounters from fiscal year (FY) 2003 to FY2015 (10/1/2002 to 9/30/2015) regardless of the date in which the patients were diagnosed with diabetes.

We used an active comparator, new user design, to mitigate the risk of immortal time bias and minimize confounding due to unmeasured characteristics [28]. We used newly initiated H2-blockers or proton pump-inhibitors (H2-PPI) as an active comparator to identify statin nonusers if they were not concurrently prescribed statins. Statin users were also newly initiated on statins. We excluded patients who previously filled prescriptions of either medication class within 12 months from cohort entry.

Index date was the date of the first prescription of statins or H2-PPI in their perspective groups. Since the study data included all available encounters from FY 2003 to FY 2015 regardless when patients were diagnosed with diabetes, the index date could have preceded, coincided, or followed their diagnosis of diabetes.

### Study intervals

The study encompassed two intervals. The baseline period, which was used to describe baseline characteristics, included the year preceding the index date. The follow-up period, which was used to ascertain outcomes, started from the index date and continued until either: (1) the last available date of VA care, or (2) end of study period, (3) death, or (4) date of statin initiation in active comparators who subsequently used a statin; among this subset of patients who entered the cohort as active comparators but subsequently used a statin, the follow up period ended as active comparators (or outcomes were censored in time-to-event analysis) at date of statin initiation and were subsequently allowed entry into the cohort as statin users starting from the date of their statin initiation as a new index date for the statin user group. We excluded patients with fewer than 60 days of follow up duration from both groups since our main outcomes would be highly unlikely to be due to fewer than 60 days of statin exposure.

### Outcomes

We used a combination of International Classification of Diseases, 9th Revision, Clinical Modification [ICD-9-CM] codes and laboratory investigations to identify outcomes. As previously published, to increase specificity of chronic diseases diagnoses using ICD-9-CM codes, we required each diagnosis to be present in $\geq 2$ separate encounters [26, 29, 30].

**Primary outcomes.**   These incident outcomes occurred during follow up period but not at baseline

1. Renal disease progression composite outcome: This dichotomous outcome comprised the presence of any of the following:

a. Doubling of mean serum creatinine during the last year of follow up in comparison to mean serum creatinine during baseline.

b. Incident stage 5 chronic kidney disease (CKD): Incident decrease in mean estimated glomerular filtration rate (eGFR) during the last year of follow up to <15 mL/min/1.73m$^2$ (stage 5) [31], using The Modification of Diet in Renal Disease (MDRD) equation (S1 File) [32].

c. Incident renal replacement therapy (S1 File).

d. Incident diabetic nephropathy: As defined by the Agency for Health Research and Quality Clinical Classifications Software (AHRQ-CCS) multilevel diagnosis category 3.3.2 (S1 File) [33].

e. Incident CKD: As defined by AHRQ-CCS diagnosis categories 156 and 158. Administrative diagnostic codes for renal events have been widely used to identify kidney diseases [34, 35] and their specificity was high (95–99%) [36].

2. Incident Diabetes with ophthalmic manifestations: As defined by AHRQ-CCS multi-level category 3.3.3 [33].

3. Incident Diabetes with neurological manifestations: As defined by AHRQ-CCS multi-level category 3.3.4 [33]. Administrative codes were commonly used in identifying patients with diabetic neuropathy in clinical research and utilization studies [37–40].

Overall, administrative codes are useful for identifying diabetic complications [41–45]. The sensitivity and specificity of ICD-9 codes for diagnosing diabetes with complications were 63.6% and 98.9%, respectively [46]. Diabetic complications codes are essential components in calculation of the Charlson comorbidity index [47] and the Elixhauser comorbidity score [48] from administrative data; both these scores are widely used [49].

**Secondary outcomes.**

1. All individual components of the composite renal disease progression outcome.

2. Change in mean creatinine (mg/dL) of individual patients during the last year of follow up in comparison to baseline.

**Negative control outcomes.**   To ensure that our findings were not due to unidentified confounders [50], we used two other outcomes that should not be affected by statins: 1) Chronic obstructive pulmonary disease (COPD) and 2) Suicide (S1 File) [51, 52].

## Cohort characterization

Patients' baseline characteristics [47], Charlson Comorbidity Index [47], and cardiovascular risk [53] were defined (S1 File). We created a propensity score to match statin-users and non-users in the overall cohort at a ratio of 1:1 using 99 variables chosen *a priori*. Using the routine of Leuven and Sianesi, we performed multivariable logistic regression to estimate the propensity score and perform nearest number matching with a caliper of 0.0008 with no replacement (S1 File) [54, 55].

## Primary analysis

We compared our primary, secondary, and negative control outcomes in the propensity score matched overall cohort using conditional logistic regression.

## Secondary analyses

We compared our primary outcomes in the following prespecified cohorts (S1 File):

1. The Overall cohort: Included all eligible patients before propensity score matching.

2. Healthy cohort: Included only patients with a Charlson comorbidity index of zero at baseline.

3. Intensive cholesterol lowering statin users in comparison to nonusers in the overall cohort [2].

4. Medium-intensity cholesterol lowering statin users in comparison to nonusers in the overall cohort [2].

5. Low-intensity cholesterol lowering statin users in comparison to nonusers in the overall cohort [2].

6. Time-to-event analysis in the propensity score matched cohort: We estimated the hazard ratio (HR) in statin users in comparison to nonusers using survival regression analysis of the following outcomes: a) Incident CKD; b) Incident diabetes with ophthalmic manifestations; and c) Incident diabetes with neurological manifestations. We performed a separate regression analysis for each of these outcomes.

7. Time-to-event analysis in the propensity score matched cohort with death as a competing risk: We estimated the subhazard ratio (SHR) in statin users in comparison to nonusers using survival regression analysis using similar outcomes to previous analysis.

## Sensitivity analysis

We examined the odds of primary outcomes after excluding those who were diagnosed with incident diabetes, incident diabetic complications, or incident cardiovascular disease within 60 days of the index date. Since it is highly unlikely that statins influenced any of these outcomes within 60 days, excluding those patients further mitigated confounding by indication or residual confounding [56–58].

## Post-hoc analysis

We performed several post-hoc analyses:

1. Propensity score-matched prevalent diabetes cohort: In this analysis, we restricted analysis to subjects with prevalent diabetes at index date. We, thereafter, created a propensity score to match statin-users and nonusers in this restricted cohort at a ratio of 1:1 using the same technique used earlier. We achieved balance in between comparison groups using a caliper of 0.00002 with no replacement.

2. Ever user vs never user cohort: Excluded patients who started as active comparators and crossed over to statin users group.

3. Incident diabetes complications cohort: Excluded patients who had any component of diabetes complications at baseline.

4. Statin duration-based analysis: We stratified statin users by duration of statin use as < 3 year of statin use, or > 3 years of statin use. Each stratum of statin users was compared to nonusers for risk of each outcome in a separate logistic regression model adjusting for the propensity score and duration of follow up.

5. Survival regression analysis with death as a competing risk in the intensive cholesterol lowering statin users in comparison to nonusers in the overall cohort and adjusting for propensity score. We performed this analysis because our secondary analysis showed that this cohort had the highest risk of complications among all other cohorts.

6. Any retinopathy and its complications: Rather than using AHRQ-CCS codes, we used a different set of ICD-9-CM codes used by other researchers (S1 Table in S1 File) [30, 59, 60].

### Other statistical analysis details

Dichotomous variables were compared using $\chi^2$ and continuous variables were compared using t-test. When Kolmogorov-Smirnov test indicated unequal distribution, we used the Wilcoxon Mann-Whitney test. We performed a separate logistic regression model for each dichotomous outcome in secondary and sensitivity analyses where the outcome was the dependent variable and statin use was an independent variable adjusting for the propensity score. Statistical significance was defined as two-tailed p-values < 0.05. Statistical analyses were performed using STATA version 15 (College Station, TX). The study was approved by the VA North Texas Health Care System and Texas Tech University Health Sciences Center Institutional Review Boards, which waived informed consent since data were fully anonymized before being accessed by the investigators. The study followed Reporting of Observational Studies in Epidemiology (STROBE) guidelines.

## Results

Cohort assembly is shown in S1 File. A total of 705,774 patients fulfilled the study criteria (595,579 statin users and 110,195 active comparators); Cohort baseline characteristics are described in S1 File. We successfully matched 81,146 pairs of statin users and active comparators (nonusers) on all baseline characteristics and duration of follow-up period (approximately 4.8 ± 3.0 years) with the exception of proportions of some racial minorities and ethnicity (Table 1). Although all the cohort had diabetes by the end of the study, at baseline period not all patients were diagnosed with diabetes yet. In the propensity score-matched cohort, statin users and nonusers had similar proportions of patients diagnosed with diabetes, all diabetes complications, and similar glycemic control. Additionally, statin users and nonusers had similar utilization of glucose lowering agents, and similar creatinine and eGFR. Baseline lipid levels were higher in statin users than nonusers (Table 1). Overall, 63% of the statin prescriptions were for simvastatin, 12% for atorvastatin, 11% for rosuvastatin, 10% for pravastatin. As expected, statin users had a greater decrease in LDL-cholesterol during follow-up compared to nonusers (mean [SD] -25.2 [31.5] mg/dL in statin users and -0.9 [23.6] mg/dL in nonusers, p<0.001)–(S1 File).

### Primary analysis

Statin use was associated with increased odds of renal disease progression (OR: 1.16, 95%CI: 1.12–1.20), ophthalmic manifestations (OR: 1.35, 95%CI: 1.27–1.44), and neurological manifestations (OR: 1.19, 95%CI: 1.15–1.25); (Table 2).

Statin users also had higher odds of incident Stage 5 CKD (OR: 1.14, 95%CI: 1.03–1.28); incident renal replacement therapy (OR: 1.11, 95%CI: 1.0–1.22); incident diabetic nephropathy (OR: 1.25, 95%CI: 1.15–1.37); and incident CKD (OR: 1.20, 95%CI 1.16–1.25). There was no difference in odds of doubling mean serum creatinine.

**Table 1. Baseline characteristics of propensity score-matched statin users and active comparators.**

| | Overall Cohort | | | Diabetes Prevalent Cohort | | |
|---|---|---|---|---|---|---|
| | Statin users (n = 81,146) | Nonusers (n = 81,146) | p-value | Statin users (n = 51,370) | Nonusers (n = 51,370) | p-value |
| **Baseline characteristics included in propensity score** | | | | | | |
| Age at index date (years): mean (SD) | 60.2 (11.6) | 60.2 (11.6) | 0.82 | 61.5 (11.3) | 61.5 (11.2) | 0.80 |
| Male Gender | 77,067 (95.0) | 77,022 (95.0) | 0.61 | 49,268 (96.0) | 49,280 (95.9) | 0.85 |
| Race | | | | | | |
| Caucasian | 55,174 (68.0) | 55,498 (68.4) | 0.08 | 35,062 (68.3) | 35,546 (69.2) | 0.001 |
| African American | 17,367 (21.4) | 17,352 (21.4) | 0.93 | 10,626 (20.7) | 10,416 (20.3) | 0.10 |
| American Indians/Alaskan, pacific/Hawaiian | 1,626 (2.0) | 1,807 (2.2) | 0.002 | 1,084 (2.1) | 1,164 (2.3) | 0.09 |
| Asian | 702 (0.9) | 464 (0.6) | <0.001 | 453 (0.9) | 322 (0.6) | 0.001 |
| Unknown/missing | 6,277 (7.7) | 6,025 (7.4) | 0.02 | 4,145 (8.1) | 3,922 (7.6) | 0.01 |
| Ethnicity | | | | | | |
| Hispanic/Latino | 5,179 (6.4) | 5,549 (6.8) | <0.001 | 3,468 (6.8) | 3,608 (7.0) | 0.09 |
| Non-Hispanic/Latino | 71,899 (88.6) | 71,645 (88.3) | 0.05 | 45,233 (88.1) | 45,168 (87.9) | 0.53 |
| Unknown/missing | 4,068 (5.0) | 3,952 (4.9) | 0.18 | 2,669 (5.2) | 2,594 (5.1) | 0.29 |
| **Social and family history during baseline period** | | | | | | |
| Family history of cardiovascular diseases[1] | 1,008 (1.2) | 1,007 (1.2) | 0.98 | 538 (1.1) | 533 (1.0) | 0.88 |
| Smoking[2] | 15,966 (19.7) | 15,964 (19.7) | 0.99 | 9,189 (17.9) | 9,271 (18.0) | 0.51 |
| Alcohol-related disorders[3] | 7,407 (9.1) | 7,375 (9.1) | 0.78 | 4,074 (7.9) | 4,146 (8.1) | 0.41 |
| Substance-related disorders[3] | 5,512 (6.8) | 5,514 (6.8) | 0.98 | 3,064 (6.0) | 3,016 (5.9) | 0.53 |
| **Vital data during baseline period** | | | | | | |
| Mean systolic blood pressure (mmHg): mean (SD) | 135 (15) | 135 (15) | 0.88 | 135 (15) | 135 (15) | 0.19 |
| Mean diastolic blood pressure (mmHg): mean (SD) | 78 (10) | 78 (10) | 0.39 | 77.5 (9.7) | 77.5 (9.6) | 0.94 |
| Body mass index | | | | | | |
| $< 25\ kg/m^2$ | 9,584 (11.8) | 9,599 (11.8) | 0.91 | 5,598 (10.9) | 5,650 (11.0) | 0.60 |
| $25\ to < 30\ kg/m^2$ | 22,509 (27.7) | 22,286 (27.5) | 0.22 | 13,666 (26.6) | 13,681 (26.6) | 0.92 |
| $30\ to < 35\ kg/m^2$ | 20,781 (25.6) | 20,925 (25.8) | 0.41 | 13,365 (26.0) | 13,295 (25.9) | 0.62 |
| $35\ to < 40\ kg/m^2$ | 10,902 (13.4) | 10,819 (13.3) | 0.55 | 7,236 (14.1) | 7,283 (14.2) | 0.67 |
| $40\ to < 45\ kg/m^2$ | 4,104 (5.1) | 4,191 (5.2) | 0.32 | 2,845 (5.5) | 2,935 (5.7) | 0.22 |
| $\geq 45\ kg/m^2$ | 2,371 (2.9) | 2,365 (2.9) | 0.93 | 1,751 (3.4) | 1,699 (3.3) | 0.37 |
| Missing | 10,895 (13.4) | 10,961 (13.5) | 0.63 | 6,909 (13.5) | 6,827 (13.3) | 0.45 |
| **Healthcare utilization during baseline period** | | | | | | |
| Number of inpatient admissions: | | | | | | |
| mean (SD) | 1.29 (3.76) | 1.30 (3.76) | 0.79 | 1.39 (3.90) | 1.42 (3.96) | 0.07 |
| median (interquartile) | 0 (0, 0) | 0 (0, 0) | 0.58 | 0 (0, 0) | 0 (0, 0) | 0.08 |
| Number of outpatient encounters | | | | | | |
| mean (SD) | 12.0 (19.5) | 12.0 (19.7) | 0.59 | 12.0 (18.3) | 11.9 (17.8) | 0.82 |
| median (interquartile) | 7 (3, 14) | 7 (3, 14) | 0.07 | 7 (3, 15) | 7 (3, 14) | 0.56 |
| Received immunization and infectious disease screening | 31,843 (39.2) | 31,846 (39.3) | 0.99 | 21,138 (41.2) | 21,206 (41.3) | 0.67 |
| Received rehabilitation care; fitting of prostheses; and adjustment of devices | 7,771 (9.6) | 7,775 (9.6) | 0.97 | 5,074 (9.9) | 5,062 (9.9) | 0.90 |
| **Diabetes and its complications during baseline period:[3]** | | | | | | |
| Diabetes mellitus | 42,242 (52.1) | 42,080 (51.9) | 0.42 | | | |
| Diabetes with complications | 9,712 (12.0) | 9,792 (12.1) | 0.54 | 10,187 (19.8) | 10,177 (19.8) | 0.94 |
| Diabetes with ketoacidosis or uncontrolled diabetes | 3,705 (4.6) | 3,697 (4.6) | 0.92 | 3,791 (7.4) | 3,820 (7.4) | 0.73 |
| Diabetes with renal manifestations | 748 (0.9) | 786 (1.0) | 0.33 | 815 (1.6) | 808 (1.6) | 0.86 |
| Diabetes with ophthalmic manifestations | 1,535 (1.9) | 1,595 (2.0) | 0.28 | 1,595 (3.1) | 1,660 (3.2) | 0.25 |
| Diabetes with neurological manifestations | 3,686 (4.5) | 3,707 (4.6) | 0.80 | 3,932 (7.7) | 3,857 (7.5) | 0.38 |

*(Continued)*

**Table 1.** (Continued)

| | Overall Cohort | | | Diabetes Prevalent Cohort | | |
|---|---|---|---|---|---|---|
| | Statin users (n = 81,146) | Nonusers (n = 81,146) | p-value | Statin users (n = 51,370) | Nonusers (n = 51,370) | p-value |
| Diabetes with circulatory manifestations | 361 (0.4) | 330 (0.4) | 0.24 | 352 (0.7) | 352 (0.7) | >0.99 |
| Diabetes with unspecified manifestations | 687 (0.9) | 713 (0.9) | 0.49 | 719 (1.4) | 753 (1.5) | 0.38 |
| Diabetic foot[4] | 469 (0.6) | 430 (0.5) | 0.19 | 434 (0.8) | 432 (0.8) | 0.95 |
| Peripheral ulcer[4] | 1,399 (1.7) | 1,409 (1.7) | 0.85 | 1,154 (2.3) | 1,131 (2.2) | 0.63 |
| Below knee amputations[4] | 7 (0.01) | 6 (0.01) | 0.78 | 2 (0.0) | 5 (0.01) | 0.26 |
| Above knee amputations[4] | 0 | 0 | n/a | 0 (0.0) | 0 (0.0) | |
| Any retinopathy & its complications[4] | 2,523 (3.1) | 2,538 (3.1) | 0.83 | 2,376 (4.6) | 2,428 (4.7) | 0.44 |
| **Glycemic control at baseline** | | | | | | |
| Mean glucose in blood in mg/dL: mean (SD) | 133 (49) | 133 (51) | 0.22 | 149 (56) | 150 (57) | 0.62 |
| At least one blood glucose of 200mg/dL or more | 16,230 (20.0) | 16,090 (19.8) | 0.38 | 16,689 (32.5) | 16,853 (32.8) | 0.28 |
| More than 5 measurements with blood glucose of 200mg/dL or more | 3,811 (4.7) | 3,814 (4.7) | 0.97 | 3,916 (7.6) | 4,016 (7.8) | 0.24 |
| **Glucose lowering medication classes at baseline** | | | | | | |
| Metformin | 15,397 (19.0) | 15,237 (18.8) | 0.31 | 15,777 (30.1) | 15,598 (30.4) | 0.26 |
| Sulphonylurea | 11,041 (13.6) | 10,965 (13.5) | 0.58 | 11,444 (22.3) | 11,329 (22.1) | 0.39 |
| GLP1 | 24 (0.03) | 14 (0.02) | 0.11 | 14 (0.03) | 17 (0.03) | 0.59 |
| DDP4 | 82 (0.1) | 88 (0.1) | 0.65 | 89 (0.2) | 91 (0.2) | 0.88 |
| Thiazolidinediones | 1,212 (1.5) | 1,202 (1.5) | 0.84 | 1,179 (2.3) | 1,247 (2.4) | 0.16 |
| α-glucosidase inhibitors | 1 (0.0) | 1 (0.0) | >0.99 | 1 (0.0) | 1 (0.0) | >0.99 |
| Amylin analog | 2 (0.0) | 3 (0.0) | 0.66 | 3 (0.0) | 3 (0.0) | >0.99 |
| SGLT2 | 1 (0.0) | 0 (0.0) | 0.32 | 0 (0.0) | 0 (0.0) | |
| Insulins | 6,674 (8.2) | 6,632 (8.2) | 0.70 | 6,721 (13.1) | 6,824 (13.3) | 0.34 |
| Total number of anti-diabetes medication groups: | | | | | | |
| mean (SD) | 0.42 (0.74) | 0.42 (0.74) | 0.33 | 0.69 (0.85) | 0.68 (0.84) | 0.66 |
| Median (interquartile) | 0 (0, 1) | 0 (0, 1) | 0.10 | 0 (0, 1) | 0 (0, 1) | 0.75 |
| **Other comorbidities at baseline[3]** | | | | | | |
| Obesity as defined by ICD-9 codes[5] | 18,621 (23.0) | 18,512 (22.8) | 0.52 | 12,801 (25.0) | 12,869 (25.1) | 0.62 |
| Valvular heart disease | 2,299 (2.8) | 2,233 (2.8) | 0.32 | 1,514 (3.0) | 1,468 (2.9) | 0.39 |
| Peri-; endo-; and myocarditis; cardiomyopathy | 1,060 (1.3) | 1,037 (1.3) | 0.61 | 703 (1.4) | 727 (1.4) | 0.52 |
| Hypertension | 52,468 (64.7) | 52,309 (64.5) | 0.41 | 35,713 (69.5) | 35,723 (69.5) | 0.95 |
| Hypertension with complication or secondary hypertension | 1,763 (2.2) | 1,798 (2.2) | 0.55 | 1,351 (2.6) | 1,346 (2.6) | 0.92 |
| Acute myocardial infarction | 273 (0.3) | 246 (0.3) | 0.24 | 182 (0.4) | 183 (0.4) | 0.96 |
| Coronary atherosclerosis and other heart disease | 9,947 (12.3) | 9,916 (12.2) | 0.81 | 7,235 (14.1) | 7,336 (14.3) | 0.37 |
| Nonspecific chest pain | 5,671 (7.0) | 5,659 (7.0) | 0.91 | 3,297 (6.4) | 3,394 (6.6) | 0.22 |
| Pulmonary heart disease | 686 (0.9) | 714 (0.9) | 0.45 | 482 (0.9) | 495 (1.0) | 0.68 |
| Other and ill-defined heart disease | 1,323 (1.6) | 1,324 (1.6) | 0.98 | 856 (1.7) | 832 (1.6) | 0.56 |
| Conduction disorders | 1,565 (1.9) | 1,607 (2.0) | 0.45 | 1,177 (2.3) | 1,179 (2.3) | 0.97 |
| Cardiac dysrhythmias | 6,842 (8.4) | 6,704 (8.3) | 0.22 | 4,631 (9.0) | 4,614 (9.0) | 0.85 |
| Cardiac arrest and ventricular fibrillation | 36 (0.04) | 34 (0.05) | 0.81 | 23 (0.04) | 31 (0.04) | 0.28 |
| Congestive heart failure | 3,399 (4.2) | 3,306 (4.1) | 0.25 | 2,485 (4.8) | 2,497 (5.0) | 0.86 |
| Acute cerebrovascular disease | 1,873 (2.3) | 1,781 (2.2) | 0.12 | 1,242 (2.4) | 1,240 (2.4) | 0.97 |
| Occlusion or stenosis of precerebral arteries; ill-defined cerebrovascular disease; Transient cerebral ischemia | 1,278 (1.6) | 1,245 (1.5) | 0.51 | 835 (1.6) | 824 (1.6) | 0.79 |
| Peripheral and visceral atherosclerosis | 2,606 (3.2) | 2,620 (3.2) | 0.84 | 1,859 (3.6) | 1,869 (3.6) | 0.87 |
| Aortic; peripheral; and visceral artery aneurysms | 725 (0.9) | 691 (0.9) | 0.36 | 456 (0.9) | 474 (0.9) | 0.55 |
| Aortic and peripheral arterial embolism or thrombosis | 118 (0.2) | 127 (0.2) | 0.57 | 78 (0.2) | 84 (0.2) | 0.64 |

(*Continued*)

**Table 1.** (Continued)

| | Overall Cohort | | | Diabetes Prevalent Cohort | | |
|---|---|---|---|---|---|---|
| | Statin users (n = 81,146) | Nonusers (n = 81,146) | p-value | Statin users (n = 51,370) | Nonusers (n = 51,370) | p-value |
| Chronic obstructive pulmonary disease and bronchiectasis | 9,854 (12.1) | 9,815 (12.1) | 0.77 | 5,998 (11.7) | 5,959 (11.6) | 0.70 |
| Asthma | 3,532 (4.4) | 3,614 (4.5) | 0.32 | 2,136 (4.2) | 2,083 (4.1) | 0.41 |
| Respiratory failure; insufficiency; arrest | 510 (0.6) | 537 (0.7) | 0.40 | 439 (0.9) | 447 (0.9) | 0.79 |
| Nephritis; nephrosis; renal sclerosis; Chronic kidney disease | 3,167 (3.9) | 3,211 (4.0) | 0.57 | 2,550 (5.0) | 2,585 (5.0) | 0.62 |
| Acute and unspecified renal failure | 1,725 (2.1) | 1,686 (2.1) | 0.50 | 1402 (2.7) | 1,392 (2.7) | 0.85 |
| Renal replacement therapy | 1,039 (1.3) | 1,049 (1.3) | 0.83 | 684 (1.3) | 693 (1.4) | 0.81 |
| Rheumatoid arthritis; Systemic lupus erythematosus and connective tissue disorders | 1,283 (1.6) | 1,289 (1.6) | 0.91 | 799 (1.6) | 777 (1.5) | 0.58 |
| Pathological fracture | 69 (0.1) | 75 (0.1) | 0.62 | 39 (0.1) | 50 (0.1) | 0.24 |
| Schizophrenia and other psychotic disorders | 2,766 (3.4) | 2,847 (3.5) | 0.27 | 1,563 (3.0) | 1,532 (3.0) | 0.57 |
| Suicide and intentional self-inflicted injury | 807 (1.0) | 810 (1.0) | 0.94 | 473 (0.9) | 469 (0.9) | 0.90 |
| Severe liver disease[6] | 425 (0.5) | 464 (0.6) | 0.19 | 353 (0.7) | 431 (0.8) | 0.005 |
| Malignancy[6] | 6,904 (8.5) | 6,912 (8.5) | 0.94 | 4,609 (9.0) | 4,654 (9.1) | 0.62 |
| Metastatic neoplasm[6] | 356 (0.4) | 382 (0.5) | 0.34 | 276 (0.5) | 288 (0.6) | 0.61 |
| Acquired Immunodeficiency Syndrome[6] | 476 (0.6) | 508 (0.6) | 0.31 | 254 (0.5) | 252 (0.5) | 0.93 |
| Any neuropathy[4] | 7,746 (9.6) | 7,799 (9.6) | 0.66 | 6,653 (13.0) | 6,569 (12.8) | 0.43 |
| **Comorbidity Scores** | | | | | | |
| Charlson Comorbidity Total Score[7]: | | | | | | |
| mean (SD) | 1.28 (1.42) | 1.29 (1.43) | 0.60 | 1.68 (1.42) | 1.69 (1.43) | 0.16 |
| median (interquartile) | 1 (0, 2) | 1 (0, 2) | 0.99 | 1 (1, 2) | 1 (1, 2) | 0.21 |
| Cardiovascular risk[8] | | | | | | |
| < 5% | 19,019 (23.4) | 19,156 (23.6) | 0.42 | 7,215 (14.1) | 7,125 (13.9) | 0.42 |
| 5 to <10% | 15,343 (18.9) | 15,379 (19.0) | 0.82 | 8,721 (17.0) | 8,692 (16.9) | 0.81 |
| 10 to <15% | 18,566 (22.9) | 18,560 (22.9) | 0.97 | 12,697 (24.7) | 12,778 (24.9) | 0.56 |
| 15 to <20% | 15,748 (19.4) | 15,647 (19.3) | 0.53 | 12,494 (24.3) | 12,609 (24.6) | 0.40 |
| 20 to <25% | 7,511 (9.3) | 7,474 (9.2) | 0.75 | 6,836 (13.3) | 6,787 (13.2) | 0.65 |
| 25 to <30% | 1,753 (2.2) | 1,722 (2.1) | 0.60 | 1,646 (3.2) | 1,651 (3.2) | 0.93 |
| ≥30% | 148 (0.2) | 139 (0.2) | 0.60 | 124 (0.2) | 136 (0.3) | 0.46 |
| Missing | 3,058 (3.8) | 3,069 (3.8) | 0.89 | 1,637 (3.2) | 1,592 (3.1) | 0.42 |
| **Renal Function at baseline** | | | | | | |
| Mean serum creatinine in mg/dL: mean (SD) | 1.11 (0.59) | 1.11 (0.61) | 0.30 | 1.12 (0.63) | 1.12 (0.63) | 0.98 |
| Mean eGFR | | | | | | |
| >90 mL/min per 1.73 m$^2$ | 21,416 (26.4) | 21,528 (26.5) | 0.53 | 14,029 (27.3) | 14,105 (27.5) | 0.60 |
| 60 to 89 mL/min per 1.73 m$^2$ | 45,065 (55.5) | 44,797 (55.2) | 0.18 | 26,927 (52.4) | 26,900 (52.4) | 0.87 |
| 45 to 59 mL/min per 1.73 m$^2$ | 10,012 (12.3) | 10,135 (12.5) | 0.35 | 6,823 (13.3) | 6,743 (13.1) | 0.46 |
| 30 to 44 mL/min per 1.73 m$^2$ | 3,266 (4.0) | 3,258 (4.0) | 0.92 | 2,513 (4.9) | 2,531 (4.9) | 0.80 |
| 15 to 29 mL/min per 1.73 m$^2$ | 944 (1.2) | 935 (1.2) | 0.84 | 718 (1.4) | 726 (1.4) | 0.83 |
| <15 mL/min per 1.73 m$^2$ | 443 (0.6) | 493 (0.6) | 0.10 | 360 (0.7) | 365 (0.7) | 0.85 |
| Mean eGFR: mean (SD) | 78 (22) | 78 (23) | 0.35 | 78 (24) | 78 (24) | 0.41 |
| **Other medications groups** | | | | | | |
| ACEI | 25,970 (32.0) | 25,868 (31.9) | 0.59 | 18,920 (36.8) | 19,003 (37.0) | 0.59 |
| ARB | 4,110 (5.1) | 4,193 (5.2) | 0.35 | 3,234 (6.3) | 3,172 (6.2) | 0.42 |
| Beta-blockers | 16,259 (20.0) | 16,379 (20.2) | 0.46 | 10,681 (20.8) | 10,674 (20.8) | 0.96 |
| Non-loop diuretic | 19,733 (24.3) | 19,804 (24.4) | 0.68 | 12,490 (24.3) | 12,531 (24.4) | 0.77 |
| Loop diuretic | 5,823 (7.2) | 5,803 (7.2) | 0.85 | 4,206 (8.2) | 4,260 (8.3) | 0.54 |

(*Continued*)

**Table 1.** (Continued)

| | Overall Cohort | | | Diabetes Prevalent Cohort | | |
|---|---|---|---|---|---|---|
| | Statin users (n = 81,146) | Nonusers (n = 81,146) | p-value | Statin users (n = 51,370) | Nonusers (n = 51,370) | p-value |
| Other anti-hypertensive agents[9] | 8,911 (11.0) | 9,017 (11.1) | 0.40 | 5,638 (11.0) | 5,560 (10.8) | 0.44 |
| Anti-arrhythmic medications | 2,830 (3.5) | 2,854 (3.5) | 0.75 | 1,829 (3.6) | 1,854 (3.6) | 0.68 |
| Antithrombotic | 2,702 (3.3) | 2,663 (3.3) | 0.59 | 1,774 (3.5) | 1,762 (3.4) | 0.84 |
| Antipsychotic | 2,754 (3.4) | 2,786 (3.4) | 0.66 | 1,588 (3.1) | 1,552 (3.0) | 0.51 |
| Dopamine agonist | 617 (0.8) | 598 (0.7) | 0.58 | 414 (0.8) | 382 (0.7) | 0.26 |
| Peripheral vascular disease medications[10] | 324 (0.4) | 311 (0.4) | 0.61 | 207 (0.4) | 206 (0.4) | 0.96 |
| Anti-smoking medications | 4,060 (5.0) | 4,006 (4.9) | 0.54 | 2,382 (4.6) | 2,376 (4.6) | 0.93 |
| Non-statin lipid lowering medications | 6,832 (8.4) | 6,839 (8.4) | 0.95 | 5,018 (9.8) | 5,035 (9.8) | 0.86 |
| **Cardiovascular procedures at baseline** | | | | | | |
| Electrocardiography | 14,468 (17.8) | 14,483 (17.9) | 0.92 | 9,264 (18.0) | 9,343 (18.2) | 0.52 |
| Echocardiography | 4,344 (5.4) | 4,334 (5.3) | 0.91 | 3,014 (5.9) | 3,027 (5.9) | 0.86 |
| Stress test | 1,898 (2.3) | 1,961 (2.4) | 0.31 | 1,198 (2.3) | 1,257 (2.5) | 0.23 |
| Cardiac catheterization | 118 (0.2) | 108 (0.1) | 0.51 | 88 (0.2) | 88 (0.2) | >0.99 |
| Percutaneous coronary intervention | 54 (0.07) | 41 (0.05) | 0.18 | 39 (0.1) | 36 (0.1) | 0.73 |
| Coronary artery bypass graft surgery | 1 (0.0) | 1 (0.0) | >0.99 | 1 (0.0) | 1 (0.0) | >0.99 |
| Pacemaker/defibrillator implantation | 55 (0.1) | 68 (0.1) | 0.24 | 47 (0.1) | 47 (0.1) | 0.76 |
| Peripheral arterial revascularization procedures | 8 (0.01) | 7 (0.01) | 0.80 | 4 (0.0) | 5 (0.0) | 0.74 |
| **Duration of Follow-up in days** | 1761 (1101) | 1770 (1101) | 0.11 | 1437 (979) | 1446 (992) | 0.13 |
| **Baseline characteristics not included in the propensity score match** | | | | | | |
| Mean total cholesterol: mean (SD)[11] | 195 (44) | 174 (39) | <0.001 | 185 (43) | 170 (40) | >0.001 |
| Mean LDL-cholesterol: mean (SD) | 119 (38) | 101 (31) | <0.001 | 111 (35) | 96 (31) | >0.001 |
| Mean HDL-cholesterol: mean (SD)[12] | 43 (12) | 41 (13) | <0.001 | 42 (12) | 41 (13) | >0.001 |

Values expressed as numbers (%) unless stated otherwise

Abbreviations: ACEI: Angiotensin converting enzyme inhibitors; ARB: Angiotensin-receptor blockers; DPP-4: Dipeptidyl peptidase 4 inhibitors; eGFR: estimated glomerular filtration rate using the Modification of Diet in Renal Disease (MDRD) Study equation;[32] GLP-1: Glucagon-like peptide 1 agonists; SGLT2 = Sodium glucose cotransporter 2 inhibitors

1. Family history of cardiovascular disease was defined using ICD-9-CM codes (S1 File)

2. Smoking as defined using ICD-9-CM codes: 3051 and V1582.

3. Diagnoses & procedures as defined by the Agency for Health Research and Quality Clinical Classifications Software disease categories (AHRQ-CCS) [33].

4. Diagnosis using ICD-9 or CPT codes as defined in prior studies (S1 File).

5. Diagnosis is based on selected ICD-9-CM diagnosis codes from category 56 of AHRQ-CCS (S1 File).

6. Malignancy, metastatic neoplasm, and Acquired Immunodeficiency Syndrome were defined using Deyo et al method in calculating the Charlson comorbidity index [47].

7. The Charlson comorbidity total score was calculated using Deyo et al method [47].

8. Cardiovascular risk was calculated using D'Agostino et al method for calculating the Framingham risk score [53].

9. Other anti-hypertensive agents include α-blocker medications, clonidine, α-methyldopa, hydralazine, minoxidil, and reserpine

10. Peripheral vascular disease medications include pentoxiphylline, cilostazole, papaverine, tolazoline, cyclandelate, and ethaverine

11. Results for total cholesterol were available for only 80,718 statin users and 80,821 control subjects in the overall cohort and 51,085 statin users and 51,163 control subjects in the diabetes prevalent cohort

12. Results for HDL-cholesterol were available for only 78,111 statin users and 78,105 control subjects in the overall cohort and 49,742 statin users and 49,792 control subjects in the diabetes prevalent cohort

Negative control outcomes were similar between statin users and nonusers (Table 2); OR of chronic obstructive pulmonary diseases was 1.0 (95%CI: 0.98–1.02) and OR of suicide was 0.98 (95%CI: 0.93–1.04).

**Table 2. Risk of outcomes during follow up period in propensity score matched cohort of statin users in comparison to active comparators.**

| | PS-Overall Cohort (Primary analysis) | | | | PS-Diabetes Prevalent Cohort | | | |
|---|---|---|---|---|---|---|---|---|
| | Statin users N (%) N = 81,146 | Active comparators N (%) N = 81,146 | OR (95% CI) | P-value | Statin users N (%) N = 51,370 | Active comparators N (%) N = 51,370 | OR (95% CI) | P-value |
| **Primary outcomes** | | | | | | | | |
| Renal disease progression composite outcome | 7,692 (9.5) | 6,724 (8.3) | 1.16 (1.12–1.20) | <0.001 | 4,980 (9.7) | 4,479 (8.7) | 1.12 (1.08–1.17) | <0.001 |
| Incident Diabetes with ophthalmic manifestations | 2,149 (2.7) | 1,602 (2.0) | 1.35 (1.27–1.44) | <0.001 | 1,931 (3.8) | 1,485 (2.9) | 1.31 (1.22–1.41) | <0.001 |
| Incident Diabetes with neurological manifestations | 5,422 (6.7) | 4,582 (5.7) | 1.19 (1.15–1.25) | <0.001 | 3,766 (7.3) | 3,593 (7.0) | 1.05 (1.00–1.10) | 0.04 |
| **Secondary outcomes** | | | | | | | | |
| Components of the composite renal disease progression outcome | | | | | | | | |
| Doubling mean serum creatinine | 1,580 (2.0) | 1,520 (1.9) | 1.04 (0.97–1.12) | 0.28 | 1,143 (2.2) | 1,083 (2.1) | 1.06 (0.97–1.15) | 0.20 |
| Incident Stage 5 CKD | 729 (0.9) | 636 (0.8) | 1.14 (1.03–1.28) | 0.01 | 542 (1.1) | 464 (0.9) | 1.17 (1.03–1.33) | 0.01 |
| Incident renal replacement therapy | 805 (1.0) | 728 (0.9) | 1.11 (1.0–1.22) | <0.05 | 547 (1.1) | 473 (0.9) | 1.16 (1.02–1.31) | 0.02 |
| Incident diabetic nephropathy | 1,209 (1.5) | 967 (1.2) | 1.25 (1.15–1.37) | <0.001 | 1,018 (2.0) | 800 (1.6) | 1.28 (1.16–1.40) | <0.001 |
| Incident CKD | 6,011 (7.4) | 5,053 (6.2) | 1.20 (1.16–1.25) | <0.001 | 3,795 (7.4) | 3,248 (6.3) | 1.18 (1.13–1.24) | <0.001 |
| Change in mean creatinine (mg/dL) from the baseline period to the last year of follow up: | | | | | | | | |
| Mean (SD) | 0.069 (0.612) | 0.063 (0.602) | - | 0.05 | 0.092 (0.614) | 0.081 (0.606) | | 0.004 |
| Median (interquartile) * | 0.00 (-0.1, 0.12) | 0.00 (-0.1, 0.13) | - | 0.007 | 0.01 (-0.1, 0.13) | 0.01 (-0.1, 0.15) | | 0.02 |
| **Negative control outcome** | | | | | | | | |
| Chronic obstructive pulmonary diseases | 23,544 (29.0) | 23,604 (29.1) | 1.0 (0.98–1.02) | 0.77 | 12,732 (24.8) | 12,754 (24.8) | 1.00 (0.97–1.03) | 0.87 |
| Suicide and intentional self-inflicted injury | 2,796 (3.5) | 2,838 (3.5) | 0.98 (0.93–1.04) | 0.57 | 1,265 (2.5) | 1,283 (2.5) | 1.01 (0.94–1.10) | 0.72 |
| **Post-hoc outcome** | | | | | | | | |
| Any retinopathy & its complications | 5,079 (6.3) | 4,264 (5.3) | 1.20 (1.15–1.26) | <0.001 | 3,613 (7.0) | 4,219 (8.2) | 1.18 (1.13–1.24) | <0.001 |

CKD = Chronic kidney diseases; PS = Propensity score

* Using Wilcoxon rank-sum test

## Secondary analysis

Statin users had higher ORs of all primary outcomes that were consistent throughout all cohorts (Table 3). The propensity score matched prevalent diabetes cohort showed overall

**Table 3. Secondary analysis and sensitivity analysis comparing outcomes during follow between statin users vs active comparators.**

| | Statin users N (%) | Active comparator N (%) | Adjusted OR* (95%CI) | p-value |
|---|---|---|---|---|
| Overall Cohort (595,579 statin users and 110,195 active comparators) | | | | |
| Renal disease progression composite outcome | 78,966 (13.3) | 7,839 (7.1) | 1.17 (1.14–1.20) | <0.001 |
| Incident Diabetes with ophthalmic manifestations | 30,202 (5.1) | 1,713 (1.6) | 1.33 (1.26–1.40) | <0.001 |
| Incident Diabetes with neurological manifestations | 61,845 (10.4) | 4,969 (4.5) | 1.17 (1.13–1.20) | <0.001 |
| Any retinopathy & its complications (post-hoc outcome) | 61,160 (10.3) | 4,691 (4.3) | 1.20 (1.16–1.24) | <0.001 |
| Healthy Cohort (148, 509 statin users and 39,009 active comparators) | | | | |
| Renal disease progression composite outcome | 15,543 (10.5) | 1,944 (5.0) | 1.31 (1. 24–1.38) | <0.001 |
| Incident Diabetes with ophthalmic manifestations | 3,294 (2.2) | 183 (0.5) | 2.09 (1.79–2.44) | <0.001 |
| Incident Diabetes with neurological manifestations | 10,803 (7.3) | 1,024 (2.6) | 1.48 (1.38–1.58) | <0.001 |
| Any retinopathy & its complications (post-hoc outcome) | 7,226 (4.9) | 713 (1.8) | 1.45 (1.33–1.57) | <0.001 |
| Intensive cholesterol lowering statin users in comparison to nonusers in the overall cohort (38,823 statin users and 110,195 active comparators) | | | | |
| Renal disease progression composite outcome | 6,534 (16.8) | 7,839 (7.1) | 1.66 (1.60–1.73) | <0.001 |
| Incident Diabetes with ophthalmic manifestations | 2,358 (6.1) | 1,713 (1.6) | 1.76 (1.64–1.89) | <0.001 |
| Incident Diabetes with neurological manifestations | 4,476 (11.5) | 4,969 (4.5) | 1.34 (1.28–1.41) | <0.001 |
| Any retinopathy & its complications (post-hoc outcome) | 4,760 (12.3) | 4,691 (4.3) | 1.63 (1.55–1.70) | <0.001 |
| Medium-intensity cholesterol lowering statin users in the overall cohort (180,884 statin users and 110,195 active comparators) | | | | |
| Renal disease progression composite outcome | 25,621 (14.1) | 7,839 (7.1) | 1.30 (1.26–1.34) | <0.001 |
| Incident Diabetes with ophthalmic manifestations | 8,945 (5.0) | 1,713 (1.6) | 1.25 (1.19–1.33) | <0.001 |
| Incident Diabetes with neurological manifestations | 20,027 (11.1) | 4,969 (4.5) | 1.23 (1.19–1.28) | <0.001 |
| Any retinopathy & its complications (post-hoc outcome) | 18,673 (10.3) | 4,691 (4.3) | 1.23 (1.18–1.27) | <0.001 |
| Low-intensity cholesterol lowering statin users in comparison to nonusers in the overall cohort (375,872 statin users and 110,195 active comparators) | | | | |
| Renal disease progression composite outcome | 46,811 (12.5) | 7,839 (7.1) | 1.13 (1.09–1.16) | <0.001 |
| Incident Diabetes with ophthalmic manifestations | 18,899 (5.0) | 1,713 (1.6) | 1.43(1.36–1.51) | <0.001 |
| Incident Diabetes with neurological manifestations | 37,342 (9.9) | 4,969 (4.5) | 1.17 (1.13–1.21) | <0.001 |
| Any retinopathy & its complications (post-hoc outcome) | 37,727 (10.0) | 4,691 (4.3) | 1.26 (1.21–1.30) | <0.001 |
| **Sensitivity Analysis** | | | | |
| Overall Cohort after excluding patients with incident diabetes, diabetic complications, or cardiovascular disease within less than 60 days from index date (353,065 statin users and 77,657 active comparators) | | | | |
| Renal disease progression composite outcome | 40,115 (11.4) | 4,595 (5.9) | 1.15 (1.11–1.19) | <0.001 |
| Incident Diabetes with ophthalmic manifestations | 11,543(3.3) | 713 (0.9) | 1.28 (1.18–1.38) | <0.001 |
| Incident Diabetes with neurological manifestations | 29,237 (8.3) | 2,553 (3.3) | 1.22 (1.17–1.27) | <0.001 |
| Any retinopathy & its complications (post-hoc outcome) | 25,147 (7.1) | 2,201 (2.83) | 1.16 (1.11–1.22) | <0.001 |
| **Post-Hoc analysis** | | | | |
| Ever user vs never user cohort (543,403 statin users and 58,019 active comparators) | | | | |
| Renal disease progression composite outcome | 73,476 (13.5) | 5,787 (10.0) | 1.05 (1.01–1.08) | 0.003 |
| Incident Diabetes with ophthalmic manifestations | 28,682 (5.3) | 1,176 (2.0) | 1.48 (1.39–1.57) | <0.001 |
| Incident Diabetes with neurological manifestations | 57,057 (10.5) | 3,492 (6.0) | 1.15 (1.11–1.19) | <0.001 |
| Any retinopathy & its complications (post-hoc outcome) | 57,261 (10.5) | 3,174 (5.5) | 1.26 (1.21–1.31) | <0.001 |
| Incident diabetes complications cohort (513,125 statin users and 98,231 active comparators) | | | | |
| Renal disease progression composite outcome | 62,994 (12.3) | 6,447 (6.6) | 1.18 (1.14–1.21) | <0.001 |
| Incident Diabetes with ophthalmic manifestations | 22,236 (4.3) | 1,205 (1.2) | 1.36 (1.28–1.46) | <0.001 |
| Incident Diabetes with neurological manifestations | 51,931 (10.1) | 4,130 (4.2) | 1.19 (1.15–1.24) | <0.001 |
| Any retinopathy & its complications (post-hoc outcome) | 41,149 (8.0) | 2,965 (3.0) | 1.24 (1.20–1.30) | <0.001 |
| Statin users for < 3 years of statin use vs nonusers (172,123 statin users and110,195 active comparators) | | | | |
| Renal disease progression composite outcome | 15,101 (8.8) | 7,839 (7.1) | 1.02 (0.99–1.05)** | 0.17 |
| Incident Diabetes with ophthalmic manifestations | 4,469 (2.6) | 1,713 (1.6) | 1.04 (0.98–0.10)** | 0.16 |
| Incident Diabetes with neurological manifestations | 9,163 (5.3) | 4,969 (4.5) | 0.84 (0.81–0.88)** | <0.001 |

*(Continued)*

**Table 3.** (Continued)

|  | Statin users N (%) | Active comparator N (%) | Adjusted OR* (95%CI) | p-value |
|---|---|---|---|---|
| Any retinopathy & its complications (post-hoc outcome) | 11,273 (6.6) | 4,691 (4.3) | 1.05 (1.00–1.08)** | 0.02 |
| Statin users for > 3 years of statin use vs nonusers (423,456 statin users and 110,195 active comparators) | | | | |
| Renal disease progression composite outcome | 63,865 (15.1) | 7,839 (7.1) | 1.19 (1.16–1.22)** | <0.001 |
| Incident Diabetes with ophthalmic manifestations | 25,733 (6.1) | 1,713 (1.6) | 1.34 (0.27–1.41)** | <0.001 |
| Incident Diabetes with neurological manifestations | 52,682 (12.4) | 4,969 (4.5) | 1.25 (1.20–1.29)** | <0.001 |
| Any retinopathy & its complications (post-hoc outcome) | 49,887 (11.8) | 4,691 (4.3) | 1.19 (1.15–1.23)** | <0.001 |

* Odds ratio adjusted for propensity score except when indicated differently

** Odds ratio adjusted for propensity score and duration of follow up

results consistent with the primary analysis. The healthy cohort had the highest OR for all complications. Similarly, intensive cholesterol lowering statin users relative to nonusers also had the highest numerical OR of all diabetes complications than less intense cholesterol lowering cohorts. Sensitivity and survival analysis supported our main analysis for all outcomes (Table 4).

## Post-hoc analysis

All post-Hoc analyses had OR that were generally consistent in direction and magnitude with primary and secondary analyses (Tables 2–4). However, renal disease progression composite outcome in the ever user vs never user design had OR of 1.05, which was lower than other analyses (Table 3).

Statin duration analysis showed that statin users for < 3 years have no increased risk of the primary outcomes and may be decreased risk of incident diabetes with neurological manifestation. However, statin users for 3 years or more had increased risks of all outcomes.

**Table 4. Hazard/Subhazard ratio of outcomes in statin users in comparison to nonusers.**

| Outcome | Hazard ratio | 95% confidence interval | p-value |
|---|---|---|---|
| Secondary analysis | | | |
| Propensity score matched cohort (81,146 statin users and 81,146 nonusers) | | | |
| Incident CKD | 1.20 | 1.16–1.25 | <0.001 |
| Incident Diabetes with ophthalmic manifestations | 1.35 | 1.26–1.44 | <0.001 |
| Incident Diabetes with neurological manifestations | 1.19 | 1.15–1.24 | <0.001 |
| Propensity score matched cohort with death as a competing risk factor (81,146 statin users and 81,146 nonusers) | | | |
| Incident CKD | 1.13* | 1.09–1.17 | <0.001 |
| Incident Diabetes with ophthalmic manifestations | 1.28* | 1.20–1.37 | <0.001 |
| Incident Diabetes with neurological manifestations | 1.19* | 1.15–1.24 | <0.001 |
| Post-Hoc analysis | | | |
| Intensive cholesterol lowering statin users in comparison to nonusers in the overall cohort with death as a competing risk factor (38,823 statin users and 110,195 active comparators) | | | |
| Incident CKD | 1.57* | 1.51–1.64 | <0.001 |
| Incident Diabetes with ophthalmic manifestations | 1.71* | 1.59–1.83 | <0.001 |
| Incident Diabetes with neurological manifestations | 1.30* | 1.24–1.36 | <0.001 |

*Subhazard ratio

## Discussion

In this study of a national cohort of veterans with diabetes, statin users compared to nonusers had modest, but significantly higher risks of incident renal, ophthalmic and neurologic complications. The results were consistent across all analyses. Moreover, there was a dose-response relation to intensity of LDL-cholesterol lowering: statin users with intensive cholesterol lowering having the highest risk for all outcomes, which strengthens our confidence in these associations. The lack of associations with the negative control outcomes (COPD, suicide) adds to specificity of these findings.

Of specific interest is that patients without comorbidities (healthy cohort) had the highest increase in odd of adverse outcomes associated with statin use. For instance, OR of renal disease progression was 1.31 (vs. 1.17 in the overall cohort), that of ophthalmic manifestations was 2.09 (vs. OR = 1.33 in the overall cohort), and that of neurological manifestations was 1.48 (vs OR = 1.17 in the overall cohort). Our findings are consistent with those from a propensity score matched cohort of a healthy Tricare population (which contains both active duty soldiers and their families) who used statins as their only prescription medication [16]. In this study, OR of diabetes with complications was 2.15 in statin users compared to nonusers, but when the analysis was restricted to healthy active duty soldiers who are expected to be healthier and more physically active, the odds was even higher at 2.47 [17].

There is biological plausibility for these associations. Statins may increase the risk of diabetes microvascular complications through increasing insulin resistance [6] and inducing mitochondrial dysfunction resulting in more toxic effects of oxygen radicals [61]. Evidence from *in-vitro* studies, a Mendelian randomization study, and observational studies have demonstrated that statin therapy is associated with insulin resistance [6–8]. Insulin resistance is associated with increased risk of diabetic complications [9, 10], endothelial dysfunction, inflammation, hypercoagulability, and increased platelet reactivity [11–13]. In presence of hyperglycemia, the availability of excessive intracellular glucose for oxidization in the tricarboxylic cycle results in production of larger amounts of electrons [62, 63]. Excessive electron burden generating superoxide may lead to diabetic complications [62, 63]. Statin therapy was associated with mitochondrial dysfunction [61, 64], which may compound the effects of superoxide. Beyond these basic science findings, we recently reported, using the same population, that statin initiation was associated with increased risk of diabetes treatment escalation and hyperglycemic events [27].

Our findings are also concordant with a recent metaanalysis of RCTs reporting that statins were associated with increased risk of renal insufficiency (OR: 1.14, 95%CI: 1.01–1.28) [65]. Yet, our study findings contrast with some of the scarce studies that examined this topic. A recent study reported that diabetic polyneuropathy in patients with newly diagnosed diabetes was similar in statin users compared to nonusers [20]. However, in that cohort 39% of new statin users discontinued their statin and 45% of statin nonusers initiated statins during the follow-up period. When the investigators censored patients at time of either initiating or discontinuing statins, there was an increased risk of diabetic polyneuropathy (HR = 1.17) [20]. Another nested matched study (with a median follow-up of 2.7 years) compared the risk of diabetic microvascular complications between patients who received statins prior to being diagnosed with diabetes and statin nonusers. Statin users had lower incidence of diabetic retinopathy (HR: 0.60) and diabetic neuropathy (HR: 0.66), but not diabetic nephropathy [18]. However, this study lacked several critical baseline characteristics such as body weight, obesity, a measure for comorbidity, or critical laboratory values, which along with the short duration of follow-up raise concerns for presence of confounders. In another retrospective cohort study, statin users had a lower rate of diabetic retinopathy (HR: 0.86) [60], however, the study

**Table 5. Number needed to be exposed for one additional harm (NNEH) from this study and number needed to treat (NNT) for cardiovascular benefit from other studies.**

| | NNEH or NNT |
|---|---|
| **Overall cohort** | |
| Adverse events as projected from our propensity score matched cohort | |
| Renal disease progression composite outcome | 83 |
| Incident Diabetes with ophthalmic manifestations | 147 |
| Incident Diabetes with neurological manifestations | 99 |
| Cardiovascular benefits for patients with diabetes as projected form a metanalysis[1] | |
| Primary prevention of MACE | 35 |
| Secondary prevention of MACE | 14 |
| **Healthy cohort** | |
| Adverse events as projected from healthy Cohort | |
| Renal disease progression composite outcome | 69 |
| Incident Diabetes with ophthalmic manifestations | 185 |
| Incident Diabetes with neurological manifestations | 83 |
| Cardiovascular benefits for patients at low cardiovascular risk as projected form a metanalysis[2] | |
| Death from any cause | 239 |
| Myocardial infarction | 216 |
| Stroke | 291 |
| Revascularization | 131 |
| **Intensive lowering of choesterol** | |
| Adverse events as projected from intensive cholesterol lowering statin users in comparison to nonusers in the overall cohort | |
| Renal disease progression composite outcome | 24 |
| Incident Diabetes with ophthalmic manifestations | 85 |
| Incident Diabetes with neurological manifestations | 69 |

MACE = Major cardiovascular event; NNEH = Number needed to be exposed to cause one additional harm as calculated in previously published formula;[67] NNT = number needed to treat

Numbers in green color indicate NNT for cardiovascular benefit and numbers in red indicate NNEH for harm from adverse events.

1. Data from Background Paper for the American College of Physicians; for primary prevention, NNT for benefit is 4.3 years; for secondary prevention, NNT for benefit is 4.9 years [68]

2. Data from a metanalysis of randomized controlled trials; low cardiovascular risk was defined as an observed 10-year Framingham risk score less than 20% for cardiovascular-realted death or nonfatal myocardial infarction in the control arm [69].

excluded patients with LDL cholesterol <100 mg/dL. The study also lacked several important baseline characteristics including predictors of diabetic retinopathy such as diastolic blood pressure and glycemic control [66]. Additionally, our duration-based analysis may offer an insight into some aspects of the conflicting results in the literature since it suggests that statin use in our study was associated with risk of outcomes in those who used statins for longer duration (≥ 3 years) but not shorter duration (< 3 years). However, since our duration-based analysis was secondary post-hoc analysis, interpreting its findings should be considered exploratory indicating necessity of prospectively designed further research with this analysis defined *a priori*.

Any adverse effects of statins should be put in context of their well-demonstrated cardiovascular benefits. Table 5 compares the calculated number needed to be exposed for one additional harm (NNEH) based on the data from this study using previously published formula

[67] and number needed to treat (NNT) for cardiovascular benefit from other studies [68, 69]. We recognize that not all events are clinically equivalent, so comparing the absolute NNEH v. NNT needs to be interpreted in a larger context. For example, a disabling stroke is clearly more morbid than doubling serum creatinine. However, a retinopathy resulting in blindness can be more devastating than revascularization following angina.

Weighing the balance of risks to benefits of statins would seem to be most important in the case of primary prevention where the absolute cardiovascular benefits are more modest, so higher risks of impactful non-cardiovascular outcomes might change decision making. Unfortunately, placebo -controlled RCTs of statins for primary prevention in the general population, which exclusively enrolled patients with diabetes or intended to specifically enroll more patients with diabetes, are limited to four RCTs (S1 File) [70].

Overall, these studies were of relatively modest size (<3000 patients in any study), were of relatively short duration (2.4–4.8 years), enrolled patients with multiple risk factors (other than diabetes), minimally (if any) assessed diabetic microvascular complications, and none of them showed a benefit (or did not report) on total mortality [71–74]. Additionally, most of these studies were done in the past century where pharmacologic agents for diabetes control, blood pressure control, and smoking were different from the present era. Recently, the incidence of acute coronary events has been declining in developed countries [75] whereas the incidence of diabetes-related complications resurged [76]. As such, it is important to incorporate all available information and critically reassess the overall harm/benefit balance of statins on all outcomes. It might be possible that, especially in the lowest risk group where the absolute cardiovascular benefits are small, the subtle adverse metabolic effects associated with statin use might tip the net balance differently than currently assumed.

This study, to our knowledge, is the largest study to date that examined the association of statin use with risk of renal diseases progression, ophthalmologic, and neurological manifestations of diabetes. Several limitations are worth noting. Although we used several methodological techniques to mitigate immortal time bias, confounding by indication, and extensively described relevant baseline characteristics, residual confounding is always a concern in observational studies. This study may have underestimated the magnitude of the outcomes since a significant proportion of its population did not have diabetes at baseline, hence, their follow up may not be long enough to manifest diabetes complications. Additionally, some studies have associated use of PPI, which we used as a control group in our study, with modest increase in renal diseases, [77, 78] some neurological conditions [79], or ophthalmic conditions [80]. Though we had detailed longitudinal data on patients within the VA healthcare system, we did not have information on potential care outside the VA system. However, it is unlikely that care outside the VA would differentially affect statin users and nonusers. Finally, VA patients are predominantly males, which may limit generalization of our data; however, research shows that male VA patients have similar health characteristics as individuals with other insurance coverage suggesting greater generalizability [81].

In conclusion, we found that among patients with diabetes, statin use was associated with a modest but significant increased risk of renal, ophthalmic and neurologic manifestations. This risk was more pronounced with intensive LDL-cholesterol lowering and in healthier populations. Further research in the use of statins for primary prevention of CVD in patients with diabetes, in which renal, ophthalmic and neurologic outcomes are specifically evaluated as primary outcomes is needed to reliably assess the overall risk benefit ratio of statins in this large segment of the population. The ethos of primary prevention should be "first do no harm".

## Supporting information

**S1 File.**
(DOCX)

## Acknowledgments

### Disclaimer

The views expressed herein are those of the authors and do not reflect the official policy or position of the Department of the Army, Department of Defense, VA Administration, or the US Government. The VA Health Care System, the University of Texas Southwestern and NIDDK had no role in the design and conduct of the study; collection, management, analysis, and interpretation of the data; preparation, review, or approval of the manuscript; and decision to submit the manuscript for publication. One of the authors (IM) is an employee of the US government. This work was prepared as part of his official duties and, as such, there is no copyright to be transferred.

## Author Contributions

**Conceptualization:** Ishak A. Mansi, Ildiko Lingvay, Song Zhang, Ethan A. Halm, Carlos A. Alvarez.

**Data curation:** Ishak A. Mansi, Matheu Chansard, Carlos A. Alvarez.

**Formal analysis:** Ishak A. Mansi, Matheu Chansard, Ildiko Lingvay, Song Zhang, Ethan A. Halm, Carlos A. Alvarez.

**Investigation:** Ishak A. Mansi, Ildiko Lingvay, Carlos A. Alvarez.

**Methodology:** Ishak A. Mansi, Matheu Chansard, Ildiko Lingvay, Song Zhang, Ethan A. Halm, Carlos A. Alvarez.

**Project administration:** Ishak A. Mansi.

**Resources:** Carlos A. Alvarez.

**Software:** Ishak A. Mansi, Matheu Chansard.

**Supervision:** Ishak A. Mansi, Ethan A. Halm, Carlos A. Alvarez.

**Validation:** Ishak A. Mansi, Matheu Chansard, Carlos A. Alvarez.

**Writing – original draft:** Ishak A. Mansi.

**Writing – review & editing:** Ishak A. Mansi, Ildiko Lingvay, Song Zhang, Ethan A. Halm, Carlos A. Alvarez.

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
