## [Decision Letter · Decision Letter 0]

22 Feb 2022

PONE-D-21-36306Statins and renal disease progression, ophthalmic manifestations, and neurological manifestations in Veterans with diabetes: A retrospective cohort studyPLOS ONE

Dear Dr. Mansi:

Thank you for submitting your manuscript to PLOS ONE. After careful consideration, we feel that it has merit but does not fully meet PLOS ONE’s publication criteria as it currently stands. Therefore, we invite you to submit a revised version of the manuscript that addresses the points raised during the review process. Please respond to the points made by myself and the 2 reviewers below.  

We look forward to receiving your revised manuscript.

Kind regards,

James M Wright

Academic Editor

PLOS ONE

Journal Requirements:

Additional Editor Comments

This is a well-conducted analysis with important findings. When describing the balance between adverse and positive outcomes it is better to use the term harm/benefit balance rather than risk/benefit balance.

Reviewers' comments:

Reviewer's Responses to Questions

**Comments to the Author**

1. Is the manuscript technically sound, and do the data support the conclusions?

Reviewer #1: Yes

Reviewer #2: Yes

2. Has the statistical analysis been performed appropriately and rigorously? 

Reviewer #1: Yes

Reviewer #2: Yes

3. Have the authors made all data underlying the findings in their manuscript fully available?

Reviewer #1: No

Reviewer #2: No

4. Is the manuscript presented in an intelligible fashion and written in standard English?

Reviewer #1: Yes

Reviewer #2: Yes

5. Review Comments to the Author

Reviewer #1: In this impressive study, much attention was paid to controlling for confounding and selection bias. Table 1 shows remarkably good balance over a very large number of variables, thanks to exposure propensity score matching. Therefore, I am inclined to interpret the data in Table 2 as I would from a hypothetical randomized trial.

As a pre-caution, I imagined a hypothetical trial…

If I imagine such a trial, patients would be randomly assigned to either statins or the control group comprising H2-blockers or proton pump-inhibitors. The cohort might be stratified into those with and without a diagnosis of diabetes at baseline.

In the subgroup without diabetes at baseline, some would develop diabetes during the follow-up period. Statins are known to increase the incidence of diabetes onset, so there would be an imbalance of diabetes between the intervention and control groups in the subgroup of people free of diabetes at baseline.

Key question: Should the analysis of the subgroup without diabetes at baseline be stratified by whether or not they developed diabetes after baseline? And after such stratification, can those who never develop diabetes in the study period be dropped from the trial analysis?

End of my hypothetical trial.

Now, turning to the cohort study, lines 144 to 147 state:

“Index date was the date of the first prescription of statins or H2- PPI in their perspective groups.

Since the study data included all available encounters from FY 2003 to FY 2015 regardless when

patients were diagnosed with diabetes, the index date could have preceded [my italics] coincided, or followed their diagnosis of diabetes.”

The fact that the cohort data source is restricted to patients who eventually are diagnosed with diabetes is analogous to dropping from the hypothetical trial (after randomization) anyone who did not later develop diabetes. Does that introduce a form of confounding like stratification on an intermediate or like M-bias (controlling for an antecedent associated with both exposure and outcome)? And if it does, what is the direction of the bias? Will propensity score matching at baseline eliminate that confounding?

I’m not sufficiently familiar with the methodology literature on directed acyclic graphs to make a judgement myself whether it is OK to restrict a cohort on its outcomes. My intuition is it is more like stratification on an intermediate than M-bias, and the bias would be towards the null. My intuition is also that such a negative bias would be eliminated by propensity score matching.

I would be convinced by reference to a methodology paper showing that restricting a cohort study to those with a certain outcome only causes underestimation of a causal effect.

Alternatively, I would be convinced by a sensitivity analysis. So, I looked in the paper for results from an analysis where the cohort was stratified by baseline diabetes, or a sensitivity analysis where the cohort was restricted to those with diabetes at baseline, before starting a statin (n =42,242) or active comparator (n =42,080)

The results of the sensitivity analysis on page 23 are a step in that direction: “Overall Cohort after excluding patients with incident diabetes, diabetic complications, or cardiovascular disease within less than 60 days from index date (353,065 statin users and 77,657 active comparators)”

But I would go further and show results excluding any whose first manifestation of diabetes was after the baseline period (i.e. analyses restricted to those who would have qualified to be in the database before follow-up). I appreciate that it would reduce the analysis to testing hypotheses concerning risk factors for further manifestations of diabetes beyond the initial manifestation. (I would also want to see an additional Table 1a showing excellent balance when restricted to patients who qualified to be in the database at baseline.)

The secondary analysis of the Healthy Cohort is restricted to a portion of those excluded from the above analysis. If there is a problem with stratifying or restricting a cohort to people who have a future outcome, then it should be manifest in this subgroup analysis. If the odds ratios were lower in this subgroup, that could be explicable as biased towards the null due to the requirement that people in the control group would have diabetes in future. On the contrary, the odds ratios are a little higher than in the overall cohort, which is to be expected if the risk difference is similar (between healthy and less healthy groups) but the baseline rate of outcomes is lower (as it would be in a healthy subgroup.) (Again, a Table 1b showing excellent balance of covariates in this group would be reassuring.)

Therefore, on the reviewer form, I checked "No" in the box referring to data transparency, which might be a bit unfair. But I wanted to flag the fact that it would be more transparent to include in the Supplementary materials, Table 1a and 1b, and the sensitivity analysis I described.

In conclusion, my judgement is that the study design and analysis were very well executed, but that the paper could briefly discuss the potential for selection bias/confounding towards the null due to the minimum requirement to be in the database was one manifestation of diabetes, and that this would be expected to be eliminated by propensity score matching.

Reviewer #2: This is an interesting article that provides valuable information on potential adverse events of statin use in diabetic patients. Authors conclude that among patients with diabetes, statin use is associated with a modest but significant increased risk of renal, ophthalmic and neurologic manifestations.

It is possible that authors underestimate the real magnitude of the association due to a number of reasons, namely,

- Not all patients had diabetes at index date. In fact, only 52% of patients were diabetic at baseline. This may have contributed to underestimation of the statin effects due to lower exposure time. Probably it would have been more informative if all patients had been diagnosed with diabetes at study entry.

- Authors state that different bias may be present in this type of study. On one hand, statin use may be falsely associated with better outcomes because of healthy-user bias, and being a surrogate for higher quality of care, or better access to healthcare. Alternatively, statin use may be falsely associated with worse outcomes because of more exposure to healthcare resulting in ascertainment bias or confounding by indication. In order to address these methodological concerns, authors employed a new user design with active comparators (H2 blockers or PPIs).

However, PPIs are associated with increased risk of renal disease (1)(2)(3)(4)(5), neurological (6)(7)(8) or ophthalmic manifestations (9)(10). This means that if statins had been compared to a drug unrelated to these adverse events, observed differences would have probably been higher than reported.

- At study entry, some 8% of patients had already developed the primary endpoint. The baseline characteristics table shows that at inclusion in the study, some patients with diabetes had also renal disease (1%), ophthalmic (2%) or neurological manifestations (4.5%). For the evaluation of incident events, these patients should have been excluded since that had already developed the study outcome. Authors showed results comparing both groups in a “Healthy Cohort” as a sensitivity analysis. A higher association between statin use and renal, ophthalmic and neurological manifestations was observed.

Interestingly, there is a dose-dependent association being the incidence of adverse events much higher if intensive cholesterol lowering compared to low-moderate lowering. The article provides no information about exposure time to statins. It would have been very informative to stratify results according to statin duration.

This article offers valuable information that suggests further research on renal, ophthalmic and neurologic manifestations of statin use is warranted.

References:

1. Al-Aly Z, Maddukuri G, Xie Y. Proton Pump Inhibitors and the Kidney: Implications of Current Evidence for Clinical Practice and When and How to Deprescribe. Am J Kidney Dis [Internet]. 2020;75(4):497–507. Available from: https://doi.org/10.1053/j.ajkd.2019.07.012

2. Fontecha-Barriuso M, Martín-Sanchez D, Martinez-Moreno JM, Cardenas-Villacres D, Carrasco S, Sanchez-Niño MD, et al. Molecular pathways driving omeprazole nephrotoxicity. Redox Biol [Internet]. 2020;32(December 2019):101464. Available from: https://doi.org/10.1016/j.redox.2020.101464

3. Guedes JVM, Aquino JA, Castro TLB, De Morais FA, Baldoni AO, Belo VS, et al. Omeprazole use and risk of chronic kidney disease evolution. PLoS One. 2020;15(3):1–16.

4. Ness-Jensen E, Fossmark R. Adverse Effects of Proton Pump Inhibitors in Chronic Kidney Disease. JAMA Intern Med. 2016;176(6):868.

5. Wu B, Li D, Xu T, Luo M, He Z, Li Y. Proton pump inhibitors associated acute kidney injury and chronic kidney disease: data mining of US FDA adverse event reporting system. Sci Rep [Internet]. 2021;11(1):1–8. Available from: https://doi.org/10.1038/s41598-021-83099-y

6. Makunts T, Abagyan R. How can proton pump inhibitors damage central and peripheral nervous systems? Neural Regen Res. 2020;15(11):2041–2.

7. Makunts T, Alpatty S, Lee KC, Atayee RS, Abagyan R. Proton-pump inhibitor use is associated with a broad spectrum of neurological adverse events including impaired hearing, vision, and memory. Sci Rep [Internet]. 2019;9(1):1–10. Available from: http://dx.doi.org/10.1038/s41598-019-53622-3

8. Novotny M, Klimova B, Valis M. PPI long term use: Risk of neurological adverse events? Front Neurol. 2019;10(JAN).

9. Hanneken AM, Babai N, Thoreson WB. Oral proton pump inhibitors disrupt horizontal cell-cone feedback and enhance visual hallucinations in macular degeneration patients. Investig Ophthalmol Vis Sci. 2013;54(2):1485–9.

10. Schönhöfer PS. Wernera B, and Tröger U. Ocular damage associated with proton pump inhibitors. BMJ 1997;314:1805

6. PLOS authors have the option to publish the peer review history of their article (what does this mean?). If published, this will include your full peer review and any attached files.

Reviewer #1: No

Reviewer #2: No

---

## [Author Response · Author response to Decision Letter 0]

26 May 2022

James M Wright

Academic Editor

PLOS ONE

Re: Statins and renal disease progression, ophthalmic manifestations, and neurological manifestations in Veterans with diabetes: A retrospective cohort study (PONE-D-21-36306)

Dear Dr. Wright:

We would like to thank you and the reviewers for your time and effort in reviewing our manuscript and for your encouraging comments. We have responded to the Editor’s and reviewers’ comments as detailed below. To facilitate the review, we included the Editor’s/reviewers’ comment in italics indented paragraph, followed by our response in regular font.

We have updated our conflict-of-interest statements, which remained basically unchanged. We look forward to hearing from you.

Ishak Mansi, MD

Response to Reviewers

Journal Requirements:

Response: We followed all instructions according to the journal requirements posted online.

Important: If there are ethical or legal restrictions to sharing your data publicly, please explain these restrictions in detail. Please see our guidelines for more information on what we consider unacceptable restrictions to publicly sharing data: http://journals.plos.org/plosone/s/data-availability#loc-unacceptable-data-access-restrictions. Note that it is not acceptable for the authors to be the sole named individuals responsible for ensuring data access. We will update your Data Availability statement to reflect the information you provide in your cover letter.

Response: The data used in this research is extracted from and hosted remotely in the VA Informatics and Computing Infrastructure (VINCI). VINCI is a Health Services Research & Development (HSR&D) Resource Center that provides researchers a nationwide view of high value VA patient data, where research projects are granted access to data in VINCI (VA HSR RES 13-457, U.S. Department of Veterans Affairs – 2008; https://vincicentral.vinci.med.va.gov). In addition to data storage, VINCI includes a cluster of servers set aside for tasks like analysis and data processing. VINCI is only accessible from the VA intranet after obtaining appropriate authorization. This means that VA researchers will have access to data and the applications necessary to data management and analysis in a secure location only accessible from the VA intranet. Data in VINCI cannot be copied, transported, or printed; several safeguards are in place to prevent any data transfer outside VINCI. Additionally, both VINCI and Privacy Officers approving IRB protocols demand investigators to attest that they will not attempt to copy or transfer data outside of VINCI. However, we will be glad to share our data with investigators who obtain appropriate authorization from VINCI to access the data.

Response: We have added the ORCID ID for the corresponding author.

Response: The ethics statement in mentioned in page 12, lines 253-255, which read: “The study was approved by the VA North Texas Health Care System and Texas Tech University Health Sciences Center Institutional Review Boards, which waived informed consent since data were fully anonymized before being accessed by the investigators.”

Response: We have modified the names of the Supporting Information files to follow the Journal guidelines and we have included a caption in the last page of the manuscript to read:

Supporting Information Captions:

- S-Methods: ……………………………………………………………………….. Page 2

o CDW

o Protocol for laboratory tests and vital signs handling

o Propensity score matching details

- S1 Table. Definition of administrative codes used in the study outcomes ……… Page 4

- S2 Table. Administrative codes used in definitions of baseline characteristics … Page 6

- S3 Table. Secondary Analysis Definitions ……………………………………… Page 10

- S-Figure. Study design and cohort assembly………………………………….... Page 11

- S4 Table. Baseline characteristics of statin users and active comparators in the 

overall cohort before propensity score matching ………………………………. Page 12

- S5 Table. Comparisons of changes in vital signs and laboratory values 

in propensity score matched cohort of statin users and nonusers ……………… Page 18

- S6 Table. Summary of the major placebo controlled randomized cardiovascular

outcome trials evaluating statins in a primary prevention in patients with diabetes Page 19

Additional Editor Comments

This is a well-conducted analysis with important findings. When describing the balance between adverse and positive outcomes it is better to use the term harm/benefit balance rather than risk/benefit balance.

Response: We have replaced the term “risk/benefit” with “harm/benefit” throughout the manuscript. 

Comments to the Author

1. Is the manuscript technically sound, and do the data support the conclusions?

Reviewer #1: Yes

Reviewer #2: Yes

2. Has the statistical analysis been performed appropriately and rigorously? 

Reviewer #1: Yes

Reviewer #2: Yes

3. Have the authors made all data underlying the findings in their manuscript fully available?

Reviewer #1: No

Reviewer #2: No

4. Is the manuscript presented in an intelligible fashion and written in standard English?

Reviewer #1: Yes

Reviewer #2: Yes

Response: We thank the reviewers for their assessment to our study. We have detailed earlier the restriction by VINCI and IRB of the VA healthcare system in sharing/transferring any data outside VINCI. As we detailed earlier, we are glad to facilitate data access by other researchers who are granted VINCI access, as per VA and VINCI policies.

Reviewer #1: In this impressive study, much attention was paid to controlling for confounding and selection bias. Table 1 shows remarkably good balance over a very large number of variables, thanks to exposure propensity score matching. Therefore, I am inclined to interpret the data in Table 2 as I would from a hypothetical randomized trial.

As a pre-caution, I imagined a hypothetical trial…

If I imagine such a trial, patients would be randomly assigned to either statins or the control group comprising H2-blockers or proton pump-inhibitors. The cohort might be stratified into those with and without a diagnosis of diabetes at baseline.

In the subgroup without diabetes at baseline, some would develop diabetes during the follow-up period. Statins are known to increase the incidence of diabetes onset, so there would be an imbalance of diabetes between the intervention and control groups in the subgroup of people free of diabetes at baseline.

Key question: Should the analysis of the subgroup without diabetes at baseline be stratified by whether or not they developed diabetes after baseline? And after such stratification, can those who never develop diabetes in the study period be dropped from the trial analysis?

End of my hypothetical trial.

Now, turning to the cohort study, lines 144 to 147 state:

“Index date was the date of the first prescription of statins or H2- PPI in their perspective groups. Since the study data included all available encounters from FY 2003 to FY 2015 regardless when patients were diagnosed with diabetes, the index date could have preceded [my italics] coincided, or followed their diagnosis of diabetes.”

The fact that the cohort data source is restricted to patients who eventually are diagnosed with diabetes is analogous to dropping from the hypothetical trial (after randomization) anyone who did not later develop diabetes. Does that introduce a form of confounding like stratification on an intermediate or like M-bias (controlling for an antecedent associated with both exposure and outcome)? And if it does, what is the direction of the bias? Will propensity score matching at baseline eliminate that confounding?

I’m not sufficiently familiar with the methodology literature on directed acyclic graphs to make a judgement myself whether it is OK to restrict a cohort on its outcomes. My intuition is it is more like stratification on an intermediate than M-bias, and the bias would be towards the null. My intuition is also that such a negative bias would be eliminated by propensity score matching.

Response: We appreciate the reviewer’s thoughtfulness in contemplating a hypothetical scenario of a clinical trial simulating our study. First, we appreciate the specific interest in a sub-cohort of patients who had diabetes at the index date (rather than developed it during the follow-up period). As such, we have added an additional propensity score-matched analysis restricted to the population who was diagnosed with diabetes prior to starting the medication of interest (statin or comparator). As you will appreciate form the revised manuscript, the results were the same. Second, the outcome in our study was diabetes complications (not development of diabetes). As such, restricting the cohort to only those with diabetes is appropriate, as that is the population of interest who has the potential to develop the outcome (diabetes-related complication). Our goal was to evaluate the effect of statin use (whether was prescribed before the diagnosis of diabetes or after the diagnosis) on occurrence or progression of diabetes-related complications. 

I would be convinced by reference to a methodology paper showing that restricting a cohort study to those with a certain outcome only causes underestimation of a causal effect.

Alternatively, I would be convinced by a sensitivity analysis. So, I looked in the paper for results from an analysis where the cohort was stratified by baseline diabetes, or a sensitivity analysis where the cohort was restricted to those with diabetes at baseline, before starting a statin (n =42,242) or active comparator (n =42,080)

The results of the sensitivity analysis on page 23 are a step in that direction: “Overall Cohort after excluding patients with incident diabetes, diabetic complications, or cardiovascular disease within less than 60 days from index date (353,065 statin users and 77,657 active comparators)”

But I would go further and show results excluding any whose first manifestation of diabetes was after the baseline period (i.e. analyses restricted to those who would have qualified to be in the database before follow-up). I appreciate that it would reduce the analysis to testing hypotheses concerning risk factors for further manifestations of diabetes beyond the initial manifestation. (I would also want to see an additional Table 1a showing excellent balance when restricted to patients who qualified to be in the database at baseline.) 

The secondary analysis of the Healthy Cohort is restricted to a portion of those excluded from the above analysis. If there is a problem with stratifying or restricting a cohort to people who have a future outcome, then it should be manifest in this subgroup analysis. If the odds ratios were lower in this subgroup, that could be explicable as biased towards the null due to the requirement that people in the control group would have diabetes in future. On the contrary, the odds ratios are a little higher than in the overall cohort, which is to be expected if the risk difference is similar (between healthy and less healthy groups) but the baseline rate of outcomes is lower (as it would be in a healthy subgroup.) (Again, a Table 1b showing excellent balance of covariates in this group would be reassuring.)

Therefore, on the reviewer form, I checked "No" in the box referring to data transparency, which might be a bit unfair. But I wanted to flag the fact that it would be more transparent to include in the Supplementary materials, Table 1a and 1b, and the sensitivity analysis I described.

In conclusion, my judgement is that the study design and analysis were very well executed, but that the paper could briefly discuss the potential for selection bias/confounding towards the null due to the minimum requirement to be in the database was one manifestation of diabetes, and that this would be expected to be eliminated by propensity score matching.

Response: We agree with the reviewer that adding an analysis restricting data to those who have diabetes at baseline will strengthen the study. Therefore, we added another propensity score-matched analysis to patients with diabetes at baseline (prevalent diabetes cohort). Overall, there was very good balance of all baseline characteristics between statin users and nonusers in the propensity score-matched prevalent diabetes cohort. Overall, the OR of outcomes were generally in line with the propensity score-matched overall cohort. However, direct comparison between OR of both cohorts is difficult since the diabetes prevalent cohort had shorter mean (SD) duration of follow up than the overall cohort: 1761 (1101) vs 1437 (979) days, respectively.

In the method section (page 2, lines 238-242), we added:

“Post-Hoc analysis: We performed several post-hoc analyses:

1. Propensity score-matched prevalent diabetes cohort: In this analysis, we restricted analysis to subjects with prevalent diabetes at index date. We, thereafter, created a propensity score to match statin-users and nonusers in this restricted cohort at a ratio of 1:1 using the same technique used earlier. We achieved balance in between comparison groups using a caliper of 0.00002 with no replacement.”

In the result section, we added

We also added the baseline characteristics to table 1 in parallel with the original propensity score matched cohort. Hence, table 1 now depicts baseline characteristics of both the propensity score-matched cohorts from the overall cohort and the diabetes prevalent cohort.

We also added the outcome results of this new analysis in table 2 as below (highlighted in light blue): 

Table 2. Risk of outcomes during follow up period in propensity score matched cohort of statin users in comparison to active comparators 

 PS-Overall Cohort (Primary analysis) PS-Diabetes Prevalent Cohort

 Statin users

N (%) 

N=81,146 Active comparators

N (%)

N=81,146 OR 

(95%CI) P-value Statin users

N (%) 

N= 51,370 Active comparators

N (%)

N= 51,370 OR 

(95%CI) P-value

Primary outcomes

Renal disease progression composite outcome 7,692 (9.5) 6,724 (8.3) 1.16 (1.12-1.20) <0.001 4,980 (9.7) 4,479 (8.7) 1.12 (1.08-1.17) <0.001

Incident Diabetes with ophthalmic manifestations 2,149 (2.7) 1,602 (2.0) 1.35 (1.27-1.44) <0.001 1,931 (3.8) 1,485 (2.9) 1.31 (1.22-1.41) <0.001

Incident Diabetes with neurological manifestations 5,422 (6.7) 4,582 (5.7) 1.19 (1.15-1.25) <0.001 3,766 (7.3) 3,593 (7.0) 1.05 (1.00-1.10) 0.04

Secondary outcomes

Components of the composite renal disease progression outcome 

Doubling mean serum creatinine 1,580 (2.0) 1,520 (1.9) 1.04 (0.97-1.12) 0.28 1,143 (2.2) 1,083 (2.1) 1.06 (0.97-1.15) 0.20

Incident Stage 5 CKD 729 (0.9) 636 (0.8) 1.14 (1.03-1.28) 0.01 542 (1.1) 464 (0.9) 1.17 (1.03-1.33) 0.01

Incident renal replacement therapy 805 (1.0) 728 (0.9) 1.11 (1.0-1.22) <0.05 547 (1.1) 473 (0.9) 1.16 (1.02-1.31) 0.02

Incident diabetic nephropathy 1,209 (1.5) 967 (1.2) 1.25 (1.15-1.37) <0.001 1,018 (2.0) 800 (1.6) 1.28 (1.16-1.40) <0.001

Incident CKD 6,011 (7.4) 5,053 (6.2) 1.20 (1.16-1.25) <0.001 3,795 (7.4) 3,248 (6.3) 1.18 (1.13-1.24) <0.001

Change in mean creatinine (mg/dL) from the baseline period to the last year of follow up: 

Mean (SD) 0.069 (0.612) 0.063 (0.602) - 0.05 0.092 (0.614) 0.081 (0.606) 0.004

Median (interquartile) * 0.00

(-0.1, 0.12) 0.00

(-0.1, 0.13) -

- 0.007 0.01 (-0.1, 0.13) 0.01 (-0.1, 0.15) 0.02

Negative control outcome

Chronic obstructive pulmonary diseases 23,544 (29.0) 23,604 (29.1) 1.0 (0.98-1.02) 0.77 12,732 (24.8) 12,754 (24.8) 1.00 (0.97-1.03) 0.87

Suicide and intentional self-inflicted injury 2,796 (3.5) 2,838 (3.5) 0.98 (0.93-1.04) 0.57 1,265 (2.5) 1,283 (2.5) 1.01 (0.94-1.10) 0.72

Post-hoc outcome

Any retinopathy & its complications 5,079 (6.3) 4,264 (5.3) 1.20 (1.15-1.26) <0.001 3,613 (7.0) 4,219 (8.2) 1.18 (1.13-1.24) <0.001

CKD = Chronic kidney diseases; PS = Propensity score

* Using Wilcoxon rank-sum test

Reviewer #2: This is an interesting article that provides valuable information on potential adverse events of statin use in diabetic patients. Authors conclude that among patients with diabetes, statin use is associated with a modest but significant increased risk of renal, ophthalmic and neurologic manifestations.

It is possible that authors underestimate the real magnitude of the association due to a number of reasons, namely,

- Not all patients had diabetes at index date. In fact, only 52% of patients were diabetic at baseline. This may have contributed to underestimation of the statin effects due to lower exposure time. Probably it would have been more informative if all patients had been diagnosed with diabetes at study entry.

- Authors state that different bias may be present in this type of study. On one hand, statin use may be falsely associated with better outcomes because of healthy-user bias, and being a surrogate for higher quality of care, or better access to healthcare. Alternatively, statin use may be falsely associated with worse outcomes because of more exposure to healthcare resulting in ascertainment bias or confounding by indication. In order to address these methodological concerns, authors employed a new user design with active comparators (H2 blockers or PPIs).

However, PPIs are associated with increased risk of renal disease (1)(2)(3)(4)(5), neurological (6)(7)(8) or ophthalmic manifestations (9)(10). This means that if statins had been compared to a drug unrelated to these adverse events, observed differences would have probably been higher than reported.

Response: We appreciate the reviewer’s input. We have created an additional propensity score-matched analysis that included only patients with diabetes at baseline, as detailed earlier (diabetes prevalent cohort). Additionally, we included the reviewer’s comment in our study limitations section. Page 34, lines2-5 now reads:

“This study may have underestimated the magnitude of the outcomes since a significant proportion of its population did not have diabetes at baseline, hence, their follow up may not be long enough to manifest diabetes complications. Additionally, some studies have associated use of PPI, which we used as a control group in our study, with modest increase in renal diseases , [78, 79] some neurological conditions,[80] or ophthalmic conditions.[81] …….

- At study entry, some 8% of patients had already developed the primary endpoint. The baseline characteristics table shows that at inclusion in the study, some patients with diabetes had also renal disease (1%), ophthalmic (2%) or neurological manifestations (4.5%). For the evaluation of incident events, these patients should have been excluded since that had already developed the study outcome. Authors showed results comparing both groups in a “Healthy Cohort” as a sensitivity analysis. A higher association between statin use and renal, ophthalmic and neurological manifestations was observed.

Response: Although some patients may have experienced kidney diseases at baseline, these same patients may not have neurological or ophthalmologic manifestations of diabetes. Hence, these patients may still experience some of our outcomes. The reviewer point of view is well taken, and we could have designed the study as suggested by the reviewer. We opted to keep those patients in the study so as we have the full spectrum of patients with diabetes. We also agree with the reviewer that our sensitivity analysis that excluded patients with any diabetes complications have addressed this concern. However, to further address this concern, we have added a post-hoc analysis that excluded patients with any diabetes complications at baseline. Overall, regardless how much we divided the data or created subgroups, our results remained consistent.

In the Methods section (page 12, lines 245-246), we added:

“ 3. Incident diabetes complications cohort: Excluded patients who had any component of diabetes complications at baseline.”

The results were presented in table 3 as the followings:

Table 3. Secondary analysis and sensitivity analysis comparing outcomes during follow between statin users vs active comparators

 Statin users

N (%) Active comparator

N (%) Adjusted

OR*

(95%CI) p-value

Incident diabetes complications cohort (513,125 statin users and 98,231 active comparators) 

Renal disease progression composite outcome 62,994 (12.3) 6,447 (6.6) 1.18 (1.14-1.21) <0.001

Incident Diabetes with ophthalmic manifestations 22,236 (4.3) 1,205 (1.2) 1.36 (1.28-1.46) <0.001

Incident Diabetes with neurological manifestations 51,931 (10.1) 4,130 (4.2) 1.19 (1.15-1.24) <0.001

Any retinopathy & its complications (post-hoc outcome) 41,149 (8.0) 2,965 (3.0) 1.24 (1.20-1.30) <0.001

Interestingly, there is a dose-dependent association being the incidence of adverse events much higher if intensive cholesterol lowering compared to low-moderate lowering. The article provides no information about exposure time to statins. It would have been very informative to stratify results according to statin duration.

This article offers valuable information that suggests further research on renal, ophthalmic and neurologic manifestations of statin use is warranted.

Response: We thank the reviewer for these comments. We have added statin duration-based analysis stratifying patients with statin use duration, which was informative and open the door for further research that is designed a priori to examine effects of statin duration on diabetes compilations. 

The methods section now states in Page 12, lines 247-250:

“ 4. Statin duration-based analysis: We stratified statin users by duration of statin use as < 3 year of statin use, or > 3 years of statin use. Each stratum of statin users was compared to nonusers for risk of each outcome in a separate logistic regression model adjusting for the propensity score and duration of follow up.”

The results of these additional subgroup analyses were added to table 3 

Table 3. Secondary analysis and sensitivity analysis comparing outcomes during follow between statin users vs active comparators

 Statin users

N (%) Active comparator

N (%) Adjusted

OR*

(95%CI) p-value

Statin users for < 3 years of statin use vs nonusers (172,123 statin users and110,195 active comparators)

Renal disease progression composite outcome 15,101 (8.8) 7,839 (7.1) 1.02 (0.99-1.05)** 0.17

Incident Diabetes with ophthalmic manifestations 4,469 (2.6) 1,713 (1.6) 1.04 (0.98-0.10)** 0.16

Incident Diabetes with neurological manifestations 9,163 (5.3) 4,969 (4.5) 0.84 (0.81-0.88)** <0.001

Any retinopathy & its complications (post-hoc outcome) 11,273 (6.6) 4,691 (4.3) 1.05 (1.00-1.08)** 0.02

Statin users for > 3 years of statin use vs nonusers (423,456 statin users and 110,195 active comparators)

Renal disease progression composite outcome 63,865 (15.1) 7,839 (7.1) 1.19 (1.16-1.22)** <0.001

Incident Diabetes with ophthalmic manifestations 25,733 (6.1) 1,713 (1.6) 1.34 (0.27-1.41)** <0.001

Incident Diabetes with neurological manifestations 52,682 (12.4) 4,969 (4.5) 1.25 (1.20-1.29)** <0.001

Any retinopathy & its complications (post-hoc outcome) 49,887 (11.8) 4,691 (4.3) 1.19 (1.15-1.23)** <0.001

* Odds ratio adjusted for propensity score except when indicated differently 

** Odds ratio adjusted for propensity score and duration of follow up

We also added in the result section the following texts (page 29, last paragraph):

“Statin duration analysis showed that statin users for < 3 years have no increased risk of the primary outcomes and may be decreased risk of incident diabetes with neurological manifestation. However, statin users for 3 years or more had increased risks of all outcomes.”

In the discussion section, we added the followings in page 32, lines 369-374, we added:

“Additionally, our duration-based analysis may offer an insight into some aspects of the conflicting results in the literature since it suggests that statin use in our study was associated with risk of outcomes in those who used statins for longer duration (≥ 3 years) but not shorter duration (< 3 years). However, since our duration-based analysis was secondary post-hoc analysis, interpreting its findings should be considered exploratory indicating necessity of prospectively designed further research with this analysis defined a priori.”

---

## [Editor Report · Decision Letter 1]

2 Jun 2022

Statins and renal disease progression, ophthalmic manifestations, and neurological manifestations in Veterans with diabetes: A retrospective cohort study

PONE-D-21-36306R1

Dear Dr. Mansi,

We’re pleased to inform you that your manuscript has been judged scientifically suitable for publication and will be formally accepted for publication once it meets all outstanding technical requirements.

Kind regards,

James M Wright

Academic Editor

PLOS ONE
---

## [Editor Report · Acceptance letter]

13 Jul 2022

PONE-D-21-36306R1 

Statins and renal disease progression, ophthalmic manifestations, and neurological manifestations in Veterans with diabetes: A retrospective cohort study 

Dear Dr. Mansi:

I'm pleased to inform you that your manuscript has been deemed suitable for publication in PLOS ONE. Congratulations! Your manuscript is now with our production department. 

Kind regards, 

on behalf of

Professor James M Wright 

Academic Editor

PLOS ONE